# Wild Fish Welfare in UK Commercial Sea Fisheries: Qualitative Analysis of Stakeholder Views

**DOI:** 10.3390/ani12202756

**Published:** 2022-10-13

**Authors:** John K. Garratt, Steven P. McCulloch

**Affiliations:** Centre for Animal Welfare, University of Winchester, Winchester SO22 4NR, UK

**Keywords:** animal welfare, fish sentience, fish welfare, wild capture, sea fisheries, fish slaughter, stunning

## Abstract

**Simple Summary:**

Every year an estimated 1.5–2.7 billion wild fish are caught by UK commercial sea fishing fleets subjecting most to severe stressors. Scientific research shows that fish are sentient beings, but UK animal welfare laws do not protect commercially caught wild fish. For example, stunning fish before slaughter is rare and gutting them alive is a common practice. There are therefore major welfare issues in wild capture sea fisheries, no laws to protect the welfare of wild fish at sea, and great potential to reform practice to improve the lives of billions of sentient fish. The article investigates this through reporting findings from interviews with 18 experts with a stake in wild fish welfare. The article reviews the UK fishing industry, policy context and scientific evidence that fish are sentient, linked to stakeholder views about what fish experience during wild capture. The article then analyses industry practices, resultant welfare harms and stakeholder attitudes towards them. No participant disputed that fish are sentient and all were able to identify key stressors to fish in different stages of the capture process, together with potential mitigations. Interviews revealed that harms can be worsened by their hidden nature along with conservative industry attitudes towards brutal practices which would be unacceptable in other areas of animal food production. The article raises awareness of an under-researched welfare problem in UK wild capture sea fisheries but should have wider significance to many fishing nations.

**Abstract:**

An estimated 1.5–2.7 billion wild fish are caught by UK commercial sea fishing fleets annually. Most are subjected to severe stressors during capture; stunning is rare and gutting alive is common practice. Fish are recognised in UK law as sentient beings, but commercially caught wild fish are excluded from welfare protections. Animal welfare impacts in wild capture sea fisheries are therefore on a massive scale, with major potential for legislative and market-based reforms. Interviews were conducted with 18 experts working within the fishing and seafood industry, fisheries management, scientific research and animal/fish advocacy organisations. The findings reveal a significant welfare gap between societally acceptable ways to treat sentient animals and the reality of capture fisheries. The participants pointed to harms caused to fish throughout different stages of the capture process caused by combinations of variables from fishing gear and methods to biological, environmental and other factors, noting that all require mitigation. Interviews revealed that the nature of harms may be exacerbated by conservative attitudes towards brutal practices in the industry, driven by profit and efficiency and free from legal restraint. To address the welfare gap, stakeholders favour engagement with the industry to improve understanding of harms and to identify mutually beneficial and shared objectives through alleviating stressors to fish in the capture process. This empirical research is focused on UK wild capture sea fisheries. However, given the dearth of welfare legislation globally, it has significance for fishing nations and the many billions of wild sea fish captured each year around the world.

## 1. Introduction

The UK commercial capture sea fishing industry comprises a fleet of approximately 11,000 fishers who land over 500 thousand tonnes of fish annually from UK waters which accounts for 81% by tonnage (87% by value) of the nation’s overall landings. A further 15% (8%) is landed from EU waters, the rest coming primarily from Norwegian waters and the North East Atlantic. It has been estimated that these figures translate to the capture of 1.5–2.7 billion wild fish each year of which the most numerous are the pelagic finfish Atlantic Mackerel (*Scomber scombrus*) and Atlantic Herring (*Clupea harengus*) [1]. EU and Norwegian boats also operate in UK waters (landing an additional 700 thousand tonnes and 200 thousand tonnes, respectively) [2]. To put these numbers in context, in the UK just over one billion land farmed animals are slaughtered annually, with over 95% of these being meat chickens [3], while approximately 200 thousand tonnes of fish are farmed and killed each year [4]. Hence, the number of wild-caught fish by UK fishers in UK waters alone is around two times the number of all land farmed animals and two to three-fold the number produced in aquaculture.

Scientific research provides good evidence that fish are sentient animals [5,6,7]. Most fish at sea are captured by nets with the largest pelagic hauls catching up to 500 thousand individual fish in one haul. Welfare harms can occur at all stages of the capture process through key stressors such as exhaustion, injury and rapid changes of environmental conditions [8]. Commercially caught wild fish commonly die during capture and those that are alive when landed are very rarely stunned prior to slaughter. More often they are left to asphyxiate or die during further processing such as gutting while still conscious [9]. The law in the UK and other jurisdictions recognises fish as sentient beings. For instance, the Animal Welfare (Sentience) Act 2022 recognises all fish as sentient beings in English law [10]. The Animal Welfare Act 2006 in England and Wales protects fish kept as pets [11]. The Welfare of Farmed Animals (England) Regulations 2007 protects fish farmed for food [12]. The Animals (Scientific Procedures) Act 1986 protects fish used for experimental purposes [13]. Despite this, and some conservation law in fisheries that has an indirect positive impact on the welfare of fish, there is no legislation in the UK to protect the welfare of wild fish caught at sea.

Given that welfare impact is a function of the number of animals affected, as well as the severity and duration of impacts [14,15], the total welfare impacts and potential for reform in commercial sea fisheries is substantial, and there is a moral imperative to avoid unnecessary suffering [16]. Despite this there are as yet very few if any experts in wild fish welfare related specifically to commercial capture. This research analyses the views and opinions of prospective stakeholders in this area, in other words some of those most closely engaged with and knowledgeable about the subject from a broad range of related fields. The objectives of the research are first, to provide an overview of the fishing industry, policy context, sentience and the wild-capture process; secondly, to explore prospective stakeholder beliefs about what fish experience during capture; and thirdly, to analyse industry attitudes and practices, welfare harms and attitudes towards them.

### 1.1. The UK Fishing Industry and Policy Context

The UK commercial fishing industry consists of wild-capture fisheries, aquaculture and fish processing, the former being the focus of this research and specifically sea fisheries. It is a devolved policy area—meaning that in the UK policy on commercial fisheries is made in England, Wales, Scotland and Northern Ireland—which accounts for 0.03% of UK total GDP. Under-10metre vessels make up 79% of the UK fleet but account for only 6% of total catch by weight, indicating the big business reality of the industry. The UK is a net importer of fish and while it is a declining sector—the UK fleet has reduced by 33% since 1996—it still lands some 600–700 thousand tonnes of fish annually worth around £1 billion, most of which is landed abroad or exported [17]. The estimated 1.5–2.7 billion wild fish caught by the UK fleet annually is based on reported capture in UK waters only [18]. There is no data on the unreported numbers which encounter capture stressors and die prematurely as a result, which are significantly higher due to Illegal, Unreported and Unregulated fishing (IUU), unrecorded discards (discards being unwanted fish returned to the water whether dead or alive), escapees and ghost fishing (the inadvertent capture of fish by lost or abandoned fishing nets or traps).

Following Brexit the UK is an independent coastal state having full responsibility for its Exclusive Economic Zone and for setting Total Allowable Catch for maximum sustainable yield of target species within UK waters. Quotas are agreed between the UK’s four nations in a fisheries Concordat and licenses distributed to fishing vessels by Devolved Administrations (Marine Management Organisation in England). Quota can legally be bought, leased or borrowed to or from other licensed operators [17], while EU fishers continue to fish in UK waters under the Trade and Cooperation Agreement (Dec 2020). There is no law protecting wild fish in UK capture fisheries. The Fisheries Act 2020 replaced the EU Common Fisheries Policy (CFP) for the sustainable management of fisheries without any provisions for fish welfare [19,20]. While the welfare of the fish is not currently considered, arguably it could be consistent with and supportive of seven of the nine specified fisheries objectives, namely: sustainability; precautionary; ecosystem; scientific evidence; bycatch; national benefit; and climate change. Some new policies have secondary welfare benefits, for example banning shark fin exports [21], and the UK is a strong proponent of protected areas for conservation and sustainability. For instance the Blue Belt Programme for Overseas Territories [22]; the High Ambition Coalition to protect 30% of oceans by 2030 [23]; and trialing five Highly Protected Marine Areas free from commercial fishing [24]. However, none of these initiatives consider fish welfare and the reality of protected areas is that capture and welfare harms are simply displaced. Wild fish and sea fishing remain exempt from the Animal Welfare Act [11]. There is no public body to provide advice on the welfare of wild animals [16]. However, the Animal Welfare (Sentience) Act 2022 will establish an Animal Sentience Committee to scrutinise Government policy for adverse impacts on all sentient animals, including wild animals [25]. This has the potential to have significant impact on the welfare of sentient beings, including fish [26,27].

### 1.2. Fish Sentience and Societal (Un)Consciousness

It is 20 years since the demonstration of nociception and associated pain like behavioural responses to noxious substances in rainbow trout [28,29,30]. Broom [6] (p. 119), concluded that the evidence shows fish to be sentient and beyond that ‘it is logical to conclude that fish feel fear and pain’ [31]. Sneddon and Brown used empirical studies to prove that fish meet Broom’s 2014 recognised definition for sentient animals, noting that the research over the last few decades has shown that fish have mental capacities on par with most other vertebrates [7]. For example, Bshary et al., (2002) found that fish cognitive abilities compare favourably with primates [32]. There is a large body of evidence indicating that fish experience pain as a negative state [5,33,34,35,36,37]. Studies show how fish react differently to painful vice non-painful stimuli and how analgesic medication changes the response. Additionally fish are motivated to avoid locations associated with a painful experience and painful events can overwhelm their normal fear or antipredator responses [38]. Pain sceptics argue that we cannot know whether fish feel pain [39], but have not produced empirical evidence to refute the hypothesis that they are likely to [38]. Furthermore studies from aquaculture demonstrate the relationship of stress to reduced meat quality and shelf-life [40] and the same relationship applies equally to wild-caught fish [41].

Sentience is generally considered to underpin the moral case to protect fish welfare [42], while the enshrining of sentience in UK domestic law means that new legislation will have to take account of all vertebrates’ capacity to experience pain and other feelings [25]. EU polling showed that 65% of those surveyed believed fish to be sentient with 79% wanting protection of their welfare and fish welfare product labelling [43]. However, animal advocacy organisations deem that the public generally view fish as less sentient than mammals, leading to a drive from them for better consumer education, for instance Compassion in World Farming’s ‘rethink fish’ campaign [44]. Traditional perceptions of fish (e.g., cold blooded, unfeeling, unevolved) do not engender public concern or empathy in the same way as for terrestrial animals thereby easing acceptance of traditional capture techniques without consciousness of the stressors fish will endure. Although out of scope here, low awareness means that welfare challenges also apply to the welfare of farmed fish in different production systems [45]. Poor consumer understanding is exacerbated by the terminology of harvest and tonnes which hide both the nature and scale of individual animal suffering, often perpetuated by media portrayal of commercial fishing as traditional and harmless. Industry practitioners may be habituated and de-sensitised to their own handling and slaughter of enormous numbers of sentient animals and there are likely to be psychological reasons why fishers might want to block out, downplay or deny fish sentience.

### 1.3. Capture Methods and Fish Welfare during Capture

According to the Food and Agricultural Organisation of the United Nations (FAO) there are 58 different fishing methods in 11 categories [46]. They are both active and passive, and in essence will either surround, herd or entice fish onto hooks (active), or entangle them in static nets or entice them into pots and other traps (passive). Common examples include:Surround—purse seine, where a wall of netting (floated at the top) is deployed from a seiner vessel to encircle pelagic species, then drawn together at the bottom like a drawstring purse thereby trapping the fish which are often subsequently pumped onboard. Common harms are caused by contact injuries along with suffocation and asphyxia from net crowding;Herd—demersal trawling (e.g., bottom trawling), where a trawl net is towed from a vessel’s bow or stern, targeting demersal fish on or near the seabed. Non-mobile species are swept up and risk contact damage and crushing while active species attempt to swim out causing exhaustion, before being confined in the cod end with further risk of asphyxiation;Hook—long lining, where horizontal or vertical lines are towed, each up to 100 km in length with additional short lines carrying baited hooks attached at intervals. Damage includes physical injury, exhaustion and risk of predation.Entangle—gillnet, a single wall of static netting invisible to the fish which on swimming into the net become entangled when the mesh is caught behind their gills. Efforts to escape cause exhaustion, contact damage and sometimes suffocation. Lost or discarded nets may continue ghost fishing.Entice—traps, set singly on the seabed or in strings with marker buoys at each end, where fish are guided through funnels that encourage entry (usually by bait) then limit escape. Less harmful than other methods though injuries can be caused through contact with the pot or other captured animals and may continue ghost fishing if lost.

For a more detailed description of the main wild capture fishing methods see Breen et al., (2020) [8].

Nearly all fishing methods cause high levels of stress to individual fish and often significant damage throughout the catch process, while the specific type of gear will determine loss rates (escapees) and how effectively the fish are targeted (selectivity), with bycatch (non-target species) and undersized fishes traditionally being discarded, usually already dead or dying [47]. Commercial fishing gear can be vast, especially in the larger vessels. For instance purse seine nets targeting pelagic species can stretch thousands of meters long and depths of over 200 m and gillnets can be several miles long. Duration can vary considerably. To illustrate demersal trawl tow times can range from minutes to a few hours depending on the density of the target species and the size and power of the vessel, while long lines, gillnets or traps may be left out for days.

Welfare harms occur throughout the capture period from when a fish encounters fishing gear until it is dead, escaped or discarded. For discards and escapees the impact of welfare harms from capture may endure beyond the confines of the capture process. However, a significant review funded by the UK government department responsible for fisheries and most animal health and welfare policy in England, the Department for Environment, Food and Rural Affairs (DEFRA), actually circumvents the welfare impacts on individual fish and cites bycatch, better targeting and ghost fishing as the primary welfare challenges for capture fisheries [48]. In most capture fisheries the fish die because of the way they are harvested rather than subsequent intentional slaughter [49], often suffering for long periods after retrieval on-board. For example, landed cod (*Gadus morhua*) can remain conscious for two hours in air [50], while following evisceration (the removal of innards known as gutting where death is caused by a combination of exsanguination or draining of blood and asphyxiation), survival time varies from around 20 min for pelagic species like herring and cod to 40 min for demersal species like plaice (*Pluronectus platessa*) [51].

### 1.4. Known Stressors during Capture and Recommended Mitigations

Breen et al. identify four stages of a catch and 11 key stressors which occur during the capture process [8], describing how the stressors impact fish welfare across different fishing methods and at different stages of the process, assessing total stress results from a combination of cumulative stress over time (Table 1).

Eurogroup for Animals is a pan-European organisation with 82 member organisations that aims to protect the wellbeing of as many animals as possible. In their report, *Catching Up: fish welfare in capture fisheries* [52], capture stressors were described similarly as seven hazards: physical injury; depredation; thermal shock; barotrauma; exhaustion; asphyxiation and crowding. The report makes 12 recommendations common across capture techniques including minimising towing speed and capture period and avoiding net overfill, along with adoption of a ‘Stewards of the Sea’ concept to engender more responsible fishing practices. Veldhuizen et al. reviewed injuries and mortality in capture fisheries based on 85 articles covering 150 species and found high mortality linked to the type of fishery and gear used, assessing overall that mortality of fish caught by trawls, purse seines and seines is higher than gillnets, hooks or traps [53]. The authors recommend the following general welfare improvement options: shorter and shallower trawls; lower catch density; better selectivity; more gradual recovery times; avoiding retrieval to the surface when temperatures are too high; and reducing the time of exposure to air. Mood and Brooke have proposed adaptations to gear and methods which minimise stressors, injury and mortality along with quicker processing, followed immediately by humane slaughter [18]. However, the specific type of trauma experienced by any fish is dependent on the type of fishing gear being used and at which point in a cycle the fish is captured [47], along with other variables like the environment. Detailed specific research into the welfare harms to date remains limited which calls for the development of tools to improve understanding [8].

## 2. Methodology

This research investigates the following research questions:(1)What do prospective stakeholders perceive wild fish capable of experiencing during capture?(2)What do prospective stakeholders think are the most significant welfare concerns in UK wild-capture fisheries?(3)How do UK fishing industry attitudes and practices contribute to the welfare concerns in wild-capture fisheries?

The research was inductive, using a qualitative approach within an interpretivist paradigm and utilised the first author’s experience with sea fisheries enforcement and government departments gleaned during a career in the UK Royal Navy. The research aim was to improve understanding of the issue based on analysis of expert perspectives from disparate areas in related fields with a prospective stake in wild fish welfare.

### 2.1. Data Collection and Sampling

In total 20 experts were spoken to. At the outset consultation took place with Phil Brooke, the research and education manager for ‘rethink fish’ at Compassion in World Farming and co-founding member of fishcount.org.uk, followed by a lead scientist at the UK Centre for Environment, Fisheries and Aquaculture Science (CEFAS). The aim of the consultations was to assist with identifying suitable interviewees (purposive sampling) and to help refine the project focus. Additional interviewees were sourced initially through contacts of the first author who have many years of experience with fisheries. This combined approach helped minimise the risk of researcher bias and identified 25 candidates from which 11 agreed to interview. Snowball sampling identified a further seven willing participants based on recommendations from original interviewees.

Initial scoping was followed by 15 semi-structured participant interviews with the 18 prospective stakeholders (three interviews comprised two individuals). There were five female and 13 male interviewees with ages ranging from mid-30s to early-60s. Participants were selected based on prominent expertise in their field, relevant experience and/or specific knowledge of the subject area. While the project was UK focused the research included overseas based contributors where they had expertise in fish welfare science and fish-focused protection and advocacy. Interviewees contributed on an individual basis on the condition of anonymity to incentivise frank discussion and came from seven different specialist areas to ensure diversity of background, organisational culture and opinions. The subject matter remains sensitive and further participant and organisational information is omitted to protect anonymity (Table 2).

Sampling aspiration was to recruit at least two individuals from each area and where that proved not possible (technical approaches and seafood) it was mitigated by participant knowledge of the technologies and the wide-ranging experience of the seafood expert. In addition, three non-industry participants had previous experience as ex-commercial fishers.

Semi-structured interviews focused on pre-determined research categories, using targeted and open-ended questions for free and in-depth discussion along with valid comparison of responses. A pilot study was conducted to test the interview preparations, technique and questions. An interview guide was divided into four question categories:(i)biggest welfare issues (by stages of a catch)(ii)policy and practice(iii)barriers (to reform)(iv)approaches and opportunities (for reform)

This paper reports findings from (i) and (ii). The findings from (iii) and (iv) will be reported in further research. Scoping was carried out during autumn 2020 and participant interviews were conducted between December 2020 and May 2021. Interviews ranged in length from 35 to 93 min with a mean duration of 62 min. Due to a combination of overseas locations and the COVID-19 pandemic lock downs in the UK, interviews were conducted remotely over Zoom, MS Teams and Skype.

### 2.2. Data Analysis

Interviews were digitally recorded and manually transcribed which along with the subsequent analysis helped to ensure accuracy and to immerse the researcher in a phenomenological way. Thematic analysis was used, initially grouped around the pre-determined interview framework of four categories to ensure maintenance of focus. All interviewees were posed at least one pre-planned question from each themed area with subsequent questions either pre-scripted or emergent depending on relevance and the answers given. The intent was to find significance in sub-themes emerging out of narratives based on interviewees experience and expertise. NVivo 12 software was used to assist first stage analysis in extracting and codifying key points. Second stage analysis used a manual method to re-order and refine the findings while a third stage referred back to NVivo themed areas for specific quotations to use in the report of the findings. To maintain participant anonymity while being able to contextualise, code and value their contributions, interviewee responses are identified by their specialist area along with participant number where required.

### 2.3. Ethics

Ethics approval for this research came from the University of Winchester via faculty level ethics review on 28 August 2020. Informed consent was attained from all participants prior to interview to allow recording and storing of data in accordance with GDPR.

## 3. Findings

### 3.1. Perspectives on Sentience, Pain and Suffering

All participants recognised that fish are sentient, although the government official (an ex-commercial fisher) noted: “it’s not something I give a lot of thought to…I don’t think I’d describe myself as conscious when it comes to the smaller fish” before clarifying “actually from the sentience perspective [tuna] are probably… no different to a smaller species”. Most interviewees also believed that fish experience pain and suffer:

“Fish have a nervous system, they have to feel pain. I just think as humans we ignore it... I do think fish feel things, I just think we put a barrier between ourselves and fish more so than we do with other animals”. (industry representative)

However, two academic scientists and the industry practitioner felt that emphasising pain or suffering can become a distraction when dealing with the industry. Instead they suggested that focus should remain on provable evidenced responses to stress (neuro-physiological and behavioural), including adaption traits like avoidance behaviours and self-coping mechanisms [54].

### 3.2. Stress, Product Quality and Sustainability

Studies showing the connection between certain stressors and reduced meat quality and shelf-life [40,41] indicate that better welfare or reduced stress and damage during capture can support greater sustainability. For example by reducing overall capture requirements or improving survival chances of escapees. Linking better welfare to greater sustainability offers potential benefits to industry in product quality, resilience, price premium and reputation. This was a major theme referenced by many interviewees. For instance the industry practitioner suggested that ‘a happy fish is usually a good quality fish’, while the seafood expert referenced the ‘direct impact on flesh quality’ of a fish fighting on a hook. All academic scientists saw shelf-life as a potential metric for welfare:

“A fish stressed immediately before it dies, the physiological changes can… affect the product quality to such a degree that it reduces its shelf-life which impacts sustainability far more than we see before”. (academic scientist P1)

### 3.3. Categorising Welfare Harms and Prioritisation

Some generalisations can be made regarding welfare harms, for example crowded nets cause crushing and rapid depth changes can cause barotrauma and sensory stresses. However, the amount of variables involved, from the number of different species and gear types to the life stage of the fish, the season, environment and duration of the capture process, makes any overall prioritisation of harms challenging:

“it’s so different from gear type to gear type what I would point out as the biggest problem… they’re all important… even if we took one example, catching herrings with a pelagic trawl, it’s still impossible for me to point out one”. (NGO scientist P11)

It’s the combination of the variables which determine whether each individual animal becomes overtaxed. To illustrate a flat fish like a turbot is able to cope with a lack of oxygen far better than blue tuna, and a tropical fish will be highly stressed when put on ice while a cold water trout’s nociceptors won’t respond (academic scientist P5). All interviewees recognised the key stressors and largely accepted their cumulative impact. The most cited concern across different gear types was the duration of the experience, seen as critical to the amount of stress and damage experienced. Other pertinent issues raised included: suffocation and asphyxia from net crowding, gill blocking or simply being left on deck (all areas); bycatch and ghost fishing (e.g., NGO official P3, government scientist P12); large temperature and light differentials (e.g., NGO official P2, government official); and lower survival of smaller individuals (e.g., academic scientist P5, government scientist P16). There was no consensus on where to focus mitigation of stressors. Views ranged from the level of exertion of the fish during the whole capture process (academic scientist P5), to ‘the impact we’re having in those minutes immediately before the animal is slaughtered’ (academic scientist P1).

#### 3.3.1. Specific Gear and Method Considerations

Gear type and capture methods are some of the key variables determining the harms experienced by individual fish. NGO officials and the industry practitioner stressed how fishing vessels, gear and methods are designed for efficiency and profit, not welfare considerations:

“I see methods not intended to be cruel but they are horribly inhumane based on the so called efficiencies. The way we catch them is not dictated by how cleanly and humanely we can do it, on the contrary it’s how efficiently we can get them out and into the storage units and processed for human consumption”. (NGO official P15)

For example, nets are generally filled to maximum density before being hauled which significantly increases crowding, suffocation, and physical injury early in the process. Likewise fishers will prioritise prepping the next haul and returning to port over quick slaughter or release (government scientist P12). According to the government official, unwanted catch is often not returned as quickly (or delicately) as possible in accordance with existing regulations, such as for protected species.

Gear modifications can facilitate unintentional secondary welfare benefits by mitigating some of the harms. Examples of this include: reducing material found on the sea bottom, or in the bottom sediments (benthic material) which saves fuel and reduces injuries in the nets (government official); four panel trawls which improve net stability and fish quality (industry practitioner); and pumping fish on-board into oxygenated water tanks to provide some restitution between capture and slaughter. Pumping on-board is fairly common practice in pelagic fisheries and improves the meat quality by providing some relief from crowding, asphyxiation and hypoxia (e.g., academic scientist P1, seafood expert).

Interviewees from all areas identified bottom trawling as the most damaging type of commercial fishing due to welfare (e.g., crowding, physical injury, suffocation, exhaustion), sustainability (e.g., bycatch, sea survival) and environmental (e.g., habitat damage) issues. Several participants, including NGO scientist P11 and the government official, emphasised how purse seining in pelagic fisheries reduces damage from debris and enables better selectivity but also accounts for the greatest numbers of individuals killed, critically with very little understanding of how they actually die. The industry practitioner emphasised damage from gill nets as ‘one of the worst’. Scientists, government and NGO officials noted how in mixed fisheries many fishermen will not know much about their likely catch composition (crucial for reducing bycatch), and lack of research and empirical knowledge of what happens to fish during the capture process is a major problem, including how they die, leaving gaps in understanding for categorising welfare issues and codifying best practice. In sum, all gear and methods are harmful (with the possible exemption of some traps) and there is no mechanism to catch a fish commercially at scale without harming its welfare to some degree.

#### 3.3.2. Stage Specific Issues and Slaughter

All four stages of a catch introduce various stressors which will affect species and individual fish welfare in different ways. Equally all stages caused concerns for interviewees without clear consensus on their relative impact. Some participants emphasized the final two stages once the fish are onboard (partly because it’s where the biggest improvements are possible), while others favoured prioritising the earlier stages where large (but unknown) numbers of fish die, after which welfare considerations are irrelevant. Summing up, the industry representative suggested “there’s probably actions you can take in each stage to diminish impact” while the seafood expert stressed that “irrespective of whether you chose to focus on one of the areas, there’s work needed on all of them”. Slaughter itself was emphasised as ethically significant by academic scientist P6 as the point of “direct human–animal suffering or welfare impact”, while NGO scientist P11 stressed the excessive severity and duration. Several other stakeholders expressed concern with the lack of stunning or deliberate slaughter method, for example:

“They’re [fish] usually just left on-board until they asphyxiate, there’s no stunning or slaughter methods really in wild fisheries”. (NGO official P3)

NGO official P4 among others stressed slaughter as potentially ‘the most practical to tackle’ while the industry practitioner emphasised it as having the best opportunity for results.

### 3.4. Welfare Related Attitudes and Practices within the Fishing Industry

One NGO official (P9) suggested that fishers ‘consider fish as a kind of fruit that they get from trees and… can do with it what they want’. The industry representative broadly agreed that ‘there’s very much an attitude of fish lives don’t matter’. Other NGO officials emphasised how the industry is de-sensitised to common practices which would be unacceptable and illegal if they were conducted on terrestrial animals. Examples given included gaffing alongside, where a handheld pole with a hook or spike at the end (a gaff) is used to swing into the body of a large fish and then pull it out of the water, and gutting alive once on-board (P4), along with extending the time of dying and exposure to environmental stressors by packing on ice (P7, government scientist P12). Again the industry representative corroborated ‘I don’t think a fisherman has ever said anything to me about if a fish feels pain… I just think those of us in the industry are de-sensitised to it’. Academic scientist P1 recounted arguments with fishers over ‘abhorrent’ practices like crushing live animals under foot, while fisheries specialists acknowledged that fishing is a conservative industry which largely accepts brutal practices as traditional, often without conscious awareness. For example:

“I spent five years as a fisherman, things that I couldn’t tolerate were crewmen…thwacking dog fish against the gunnel…because they were clogging up the nets… gutting [alive] is another example that’s a fairly barbaric act but necessary to preserve the quality of the fish”. (government official)

Here the interviewee means that it is necessary to gut the fish quickly to preserve the quality of the meat. However, it is not necessary to gut the fish whilst alive to preserve the quality.

The industry representative believed that for industry to acknowledge wild fish welfare, though necessary, was akin to opening a Pandora’s Box. The industry practitioner concurred: ‘I think fish welfare now is a real can of worms for the fishing industry, as soon as you open that you’ve got no idea where it’s going to end’. Despite this there are a few industry first movers pursuing on-board stunning technology (e.g., Ekofish), while fishing gear is eminently adaptable and both industry specialists emphasised that fishers are willing to try things if the conditions suit them, such as simple processes and financial incentives or other clear benefits. The key to improving practices may lie in identifying where the interests of fishers and fish coincide:

“Fish in a trawl net suffocate and get squashed and the fishermen have realised… where they trawl for a much shorter period of time… there’s more fish there for them to catch [and] they recognise there’s better quality fish…, better yields and they’re getting better prices, and that is also of benefit to the fish”. (seafood expert)

However, the government official, government scientist P16 and the industry practitioner all mentioned instances where skippers have been paid to trial new nets, had positive results, yet still subsequently reverted to their old ways.

Government scientist P16 believed that any significant and lasting change will require fishers “having a different view of what they’re seeing”. NGO official P9 expressed it as changing attitudes “to make them conscious that they are working with living animals…that can suffer”, emphasising the psychological and commercial challenges with this for fishers, “is it still possible then to catch fish in that enormous amount”. The seafood expert cited evidence of welfare improvements in aquaculture over the last 20–30 years as proof that attitudes can change with time. Additionally, academic scientist P1 explained that when Norway enacted a discard ban in 1987 it “changed the culture within the industry, it’s changed the way they think” meaning that “fisherman now consciously avoid discarding” and younger fishermen “look on the old practice as being abhorrent”. Furthermore, according to government scientist P16, big innovations like the EU Landing Obligation (2019) “raised the profile of unwanted catches and made fishermen aware of them” so that they are now more conscious of what they are doing, even if the regulations themselves were flawed (government official).

## 4. Discussion

The killing of an estimated 1.5–2.7 billion fish within UK capture fisheries every year underestimates the true scale of harms wild fish experience through capture as it omits those unquantified yet indisputably vast numbers of individual animals dying additionally, and/or experiencing extreme stressors as a by-product of the industry and its methods. Reflecting that welfare impact is a function of the numbers of animals affected, as well as the severity and duration of impacts, accepting that fish are sentient animals and noting that there are no welfare specific protections for wild fish, this indicates a welfare deficit in capture fisheries that may fail the moral imperative to avoid unnecessary suffering. The interviews exposed different attitudes towards wild fish welfare but did reveal some common ground between prospective stakeholders offering potential to address some of the concerns identified.

### 4.1. Recognition of Sentience, Suffering and the Hidden Nature of Welfare Harms

Acknowledging sentience is key for catalysing governmental, industry, and public recognition of widespread and substantial welfare harms caused to wild fish in commercial capture fisheries. The scientific view on fish sentience is unequivocal and fish have been protected in UK and EU law for many years. All participants believed that fish are sentient beings, so it is a major regulatory problem that commercially caught wild fish have no legal protection beyond the indirect sustainable management of fisheries. There are inconsistencies in the treatment of wild-caught fish, which would be condemned and punishable in law if practiced on fish and other types of sentient animals in other sectors of animal use. Jennings et al. have written how in the UK and many other nations ‘no livestock farmer or aquaculture worker could legally treat animals in the way that commercial fisherman are legally allowed to’ [38] (p. 925). Many stakeholders interviewed for this research recognised the same.

In addition, poor consumer understanding, often exacerbated by industry terminology and media portrayal, along with the concept of fish pain relating the subjective nature of others minds, continue to conceal the magnitude of suffering caused and are arguably used by some close to the industry as a distraction from acknowledging the extent of harms involved. Combined, this state of affairs means there is a large welfare gap between ways that British society believe are acceptable to treat sentient animals, and the reality of the treatment of wild-caught fish in UK capture fisheries.

### 4.2. Acknowledging Welfare Harms and Closing the Welfare Gap

Welfare harms arise due to the severe hazards and stressors inherent in commercial capture methods, which are exacerbated by traditional practices driven by business efficiency factors. Industry focus on these factors means fish are viewed as a product and neglects any consideration of them as sentient animals, with little reputational damage for doing so. Therefore, known reforms which could improve fish welfare, such as slower trawls, less full nets, shorter soak times or even just prioritising processing the catch on deck over preparing the next catch, are not commonly practiced in industry due to the perception that there are no economic or commercial benefits. Interviewees all agreed that the challenges presented by the numerous variables and the harms caused by all gear types means that even with political will, money, an educated public and a supportive industry, optimal fish welfare is impossible. Indeed, NGO official P9 stated in interview that you would have to ban commercial wild fish capture to achieve this.

However, more open industry acknowledgment of the welfare harms caused would allow a much needed discussion about the nature of the stressors being experienced by individual fish during commercial capture, and a pragmatic stakeholder dialogue about how and where to address them. This could start with areas of mutual benefit, where the interests of the fish and the fishers coincide. Furthermore, admitting the limited understanding of the detailed specifics of the harms done to wild fish during capture, including very often how they actually die, could help facilitate a drive for greater knowledge including what best practice might entail. Linking enhanced welfare to better quality product with potential price premium and sustainability advantages, along with reputational consequences for ignoring the major welfare problems, were seen by most participants as the way to start to close the welfare gap.

### 4.3. Categorising Harms

All participants agreed that more research is needed to better understand welfare harms, how they can be mitigated and what best practice looks like. Bottom trawling was cited by all stakeholders as the most destructive fishing method overall (environmental damage, poor targeting and welfare issues) but for welfare specifically it is harder to categorise relative harms. There is no objective evidence available to compare the impacts of key stressors from, for example, being churned around in a two hour active beam trawl (crowding, physical injury, suffocation, exhaustion), compared to 24 h trapped in a static gillnet (suffocation, physical injury, predation). Equally, attempting to determine which stage and specific activity during the capture process does the most harm is speculative, and an area where knowledgeable persons collectively hold non-conclusive views. All interviewees gave the impression that it is not possible to rank the experiences for individual fish and regardless to do so is of limited value when you can determine that all methods, gear and stages of a catch cause harms which require mitigation of some form.

### 4.4. Addressing Prevailing Attitudes

The fishing industry is conservative and largely de-sensitised to seeing anything wrong with the practices used, while understanding or even awareness of welfare relevant issues, from catch composition in mixed fisheries to high mortality in pelagic fisheries, is generally poor. Consumer consciousness of welfare issues in wild-caught fish is also relatively weak. Interviewees connected to industry acknowledged welfare issues could become problematic in future recognising an advantage in addressing them early. For instance, the industry representative recognised how it is ‘something [the fishing industry] need to start considering and think about sensibly, not that it’s just a vegan crusade’. However, overriding concerns about unknown consequences means industry may equally well block any attempts to address welfare issues.

Politically, Brexit and leaving the CFP have caused significant change and uncertainty, leaving few operators willing to invest in newer technologies (industry practitioner), while industry may well oppose fish welfare regulatory reform as unworkable due to the anticipated impact to livelihoods (government official). Industry also has many perceived higher priority issues, sometimes fundamental to their own commercial viability (seafood expert). However, there is evidence that attitudes can change over time with the right stimulus, for instance following Norway’s discard ban, leading to a greater consciousness of practices. In addition, government scientists spoke of education proving effective in changing attitudes in the UK scientific community and the seafood expert believed that while education is not yet widespread or institutionalised across the fishing industry, it could fairly easily be expanded to good effect.

Interviewees broadly agreed that to build trust, overcome resistance and address the issue constructively would require key industry stakeholders engaging in an open and evidence based conversation about the science and welfare conscious practices, while NGOs engage consumers and funding bodies. Noting experience from illegal fishing which showed that ‘trying to get anything done economically speaking was a tremendous barrier’ (NGO official P7), progress will require demonstration of reciprocal benefits and mutually advantageous outcomes. Focusing on reducing capture stress for better quality, which means a higher price, longer shelf-life and enhanced survival of unwanted catch was the predominant theme interviewees assessed would most likely incentivise welfare conscious behaviours. One critique by academic scientist P6 suggested that price premiums may be lost in the chain rather than helping the fish thereby risking ‘welfare washing’. However, making the link between welfare and sustainability helpfully provides a connection to the world of extant accreditation schemes too.

## 5. Conclusions

The Animal Welfare (Sentience) Act 2022 recognises all fish as sentient beings in English law. Fish welfare considerations are therefore applicable to all fish, including commercially caught wild fish at sea which currently have no direct legal protection. This research reports findings from interviews with 18 fish welfare related specialists and adds to the limited understanding around fish welfare in capture fisheries, including expert views on what fish experience during capture, the nature of welfare harms, and perceptions of industry practices and attitudes.

Fish are sentient beings that react to and are harmed by wild-capture induced stressors in ways which are consistent with pain and suffering. The categorisation and prioritisation of welfare harms is difficult due to the number of variables involved and the lack of research, good data and understanding, however all commercial sea fishing methods cause significant harms at each stage of the capture process. Interviews reveal that the nature of harms is exacerbated by conservative attitudes and often brutal practices in the fishing industry, driven by profit and efficiency, free from legal restraint. Arguably, capture fisheries methods are inconsistent with morally and socially acceptable ways to treat sentient animals, meaning that there is a welfare gap and a major regulatory problem that wild fish caught at enormous scale have no direct legal protection.

The UK fishing industry may be apprehensive about engaging in what it might perceive to be a burden and even risk to its operations. Arguably, an ongoing discussion is needed between interested academic, governmental, industry and NGO specialists to determine a way forward to begin to address this welfare gap in a way which offers some reciprocal benefits to industry. The science and UK law are in agreement on fish sentience, which is a solid basis on which to recognise that the stressors inherent in commercial capture methods do cause welfare harms to wild fish. Open acknowledgment by industry that harms are caused would allow a more detailed discussion about the nature of the stressors and could open the way to understanding what improving welfare and best practice might look like, along with any regulation required to enact it. A collaborative approach to identify common ground, reciprocal benefits and mutually agreed trials and outcomes is most likely to succeed. Linking better welfare to product quality with potential price premium and sustainability advantages, along with avoiding reputational consequences for ignoring the issue, seems like a pragmatic place to start.

This paper has focused on the welfare of wild sea fish caught in UK waters. However, the commercial practices used and the lack of any meaningful attention to welfare of wild fish and legal protections are a common feature not only in the UK but globally. Given this the magnitude of the problem is far bigger than reported here, and the findings have significance on an arguably global basis.

## Figures and Tables

**Table 1 animals-12-02756-t001:** Four stages of a catch and the stressors encountered. Adapted from Breen et al., (2020) [8].

**S T A G E S**	STRESSORS
**1. Capture**	**2. Retrieval**	**3. Handling & Sorting**	**4. Endpoint**
			**a. Slaughter**	**b. Release/Escape**
Crowding	Barotrauma	Emersion	Emersion	Temperature shock
Hypoxia	Temperature shock	Crowding	Crowding	Barotrauma
Injury	Osmoregulatory distress	Hypoxia	Temperature shock	Osmoregulatory distress
Fatigue/Exhaustion	Injury	Injury	Hypoxia	Fatigue/Exhaustion
	Light exposure	Light exposure		Emersion
	Crowding	Temperature shock		Injury
	Emersion			Light exposure
				Displacement
				Predation
Cumulative stress 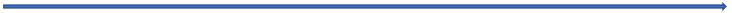

**Table 2 animals-12-02756-t002:** Interviewee decode by specialist area.

Serial	Area (Country)	Participant No.
1	Fish academic/scientist (Norway) (Netherlands)(NL)	P1 P5, P6
2	Seafood expert (United Kingdom)(UK)	P14
3	Government fisheries specialist (UK): Scientists Official (Fish/fisheries focussed executive bodies sponsored by DEFRA)	P12, P16 P13
4	Fishing industry (UK): Representative Practitioner	P17 P18
5	Humane focus NGO (UK) (United States)(US)	P2 P7, P8, P15
6	Animal advocacy NGO (European Union) (NL) (Denmark)	P4 P10 P11
7	Fish advocacy NGO (US) (NL)	P3 P9

## Data Availability

Restrictions apply to the availability of these data. Data presented in this study was obtained from interviewees on the condition of anonymity. Data may be requested from the corresponding author but will only be released with the permission of the relevant interviewee.

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
