# Peer review of "Wild Fish Welfare in UK Commercial Sea Fisheries: Qualitative Analysis of Stakeholder Views"

_animals, 2022, doi:10.3390/ani12202756_

Round 1

Reviewer 1 Report

The theme evoked by the authors is really interesting and well elaborated. Overall, I recommend the paper for publication. 

The present paper focused on the welfare of fish caught in UK waters and also those obtained in aquaculture in the UK area. In this paper the authors evaluated how the welfare of the fish can be affected by the type of fishing and fishing methods. The authors also review the different studies based on the fish stress and the fish pain. Finally, they evaluated how the stress and pain can affected the fish quality.

I consider the paper Original because firstly the type topics very little study we treat this subject whose interest is based on the welfare of the fish which could be interesting for the readers of this newspaper and the world. secondly due to the fact that the analyzes were very well done due to the different interviews carried out at several layers of the population which allows to have a varied idea.

Compared to other published material is not really original but for the subject area which is really difficult because firstly due to the critical lack of published research and secondly the manner in which the authors have used easily the material and method to assess the psychology of fish and evaluated its impact on firstly capture and quality.

Concerning the questions raised by the authors of which they have developed well, I think they should also develop the impact of fishing methods on the welfare of fish and how it can be improve, more particularly the other methods apart from the bottom trawl donated several studies are delayed. for that it should better exploit the literature concerning the study to develop this part.

as I mentioned above the conclusion reflect the objectives from the introduction and arguments present.

The references are presented according to the recommendations of the journal and all the documents cited support the arguments raised by the authors.

For the tables the authors must redo them according to the recommendations of the journal and define the acronyms for a good understanding of the readers. I also think the authors should add a flowchart at the methodology level to graphically explain the  mechanism developed.

Reviewer 2 Report

A well written manuscript on a sensitive subject, covering in a holistic way the stakeholder beliefs and attitudes towards commercial capture fisheries welfare.

I have no other comments to make except for paragraph 2.3 correct GPDR with GDPR.

Reviewer 3 Report

This is an excellent review that presents a very interesting study of fishing in UK waters. The review part is incredibly thorough. The survey outcomes are vitally important and this is a very timely topic with substantial growing interest in the welfare of wild caught fish. I have no hesitation in recommending this manuscript for publication. The writing is clear and first class. 

Reviewer 4 Report

Garrat and McCulloch (ID Manuscript animals-1910572-peer-review-v1) provided a current review of the UK legislation and policy on fish welfare, providing several scientific evidence of fish sentience and fishing industry practices. Moreover, the authors interviewed experts from a wide range of areas related to fish welfare. These interviews allowed the authors to identify the key stressors during the fish-capturing process. The general view suggested that, although fish are recognized as sentient, industry attitudes are far away from providing adequate maintenance conditions and food production process compared to the other farming species. Garrat and McCulloch aimed to raise awareness of the lack of welfare legislation in UK capture fisheries and other nations.

Garrat and McCulloch have provided a plethora of information regarding fish welfare in an easy-to-read study that can be easily understood by whom experts in this field and the general public are not like. The topic is well described and supported by previous findings, as well as the discussion of their results. The present article is a suitable contribution to the field. 

I have some minor comments for the authors.

Title: the present article is a review/current opinion of UK legislation relying on fish welfare. In addition, the authors support them by interviewing several experts on fish welfare. The term “analysis” would mainly evoke the use of an analytical approach to support some hypothesis or meta-analysis techniques for reviewing the main effects from previous studies. I would suggest the second part of the title as “review” or “qualitative analysis”. 

In the second paragraph, I will shortly mention the fact that wort maintaining conditions critically impoverish the quality of food (see for example Braithwaite, V. A., & Salvanes, A. G. V. (2010). Aquaculture and restocking: implications for conservation and welfare. Anim Welf, 19, 139-149.)

I would ask about the means of “environmental conditions” as a key stressor. I understand that authors would emphasize the poor and worst capture conditions in which fish are subjective. However, the terms “environmental conditions” evoke more the sea/ocean in which animals live. I guess “capture conditions” would be more suitable for their purpose.

Authors should report some evidence on the morphological, cognitive, and food quality consequences of pain in fish (1.1 Fish sentience and societal (un)consciousness). For example, high levels of cortisol, a natural response to stress conditions (a standard index of “pain”), affects the quality of products (rate growth or fecundity) or the possibility of survival for whom population that has been raised for repopulating purpose (see the abovementioned reference from Braithwaite and Salvanes, 2010) 

I would not so strictly define sentience as the most moral foundation to protect animal welfare. Indeed, not all species have been provided to be sentient (i.e., sponges and jellyfish), or even those who are defined as sentience (i.e., invertebrates) are not protected as well as vertebrate species.

Fish farming represents the largest captive breeding activity of vertebrates and continues to increase exponentially as natural populations of fish are in marked decline. However, there is still low awareness of maintaining fish in high-level welfare conditions even in the aquaculture field (see an example: Macaulay, G., Bui, S., Oppedal, F., & Dempster, T. (2021). Challenges and benefits of applying fish behavior to improve production and welfare in industrial aquaculture. Reviews in Aquaculture, 13(2), 934-948.). The authors would briefly discuss that the scarcity of fish welfare awareness seems to be present in several areas of food farming.
